# A Modified Robotic Manipulator Controller Based on Bernstein-Kantorovich-Stancu Operator

**DOI:** 10.3390/mi14010044

**Published:** 2022-12-24

**Authors:** Qianqian Zhang, Mingye Mu, Xingyu Wang

**Affiliations:** 1Philips Health Technology (China) Co., Ltd. Shenyang Branch, Shanghai 200233, China; 2Technical Inspection Center, Shenyang Aircraft Corporation, Shenyang 110850, China; 3Marine Equipment & Technol Inst, Jiangsu University of Science and Technology, Zhenjiang 212000, China

**Keywords:** robotic manipulator, Bernstein–Kantorovich–Stancu operator, full order observer, end effectors, robotics

## Abstract

With the development of intelligent manufacturing and mechatronics, robotic manipulators are used more widely. There are complex noises and external disturbances in many application cases that affect the control accuracy of the manipulator servo system. On the basis of previous research, this paper improves the manipulator controller, introduces the Bernstein–Kantorovich–Stancu (BKS) operator, and proposes a modified robotic manipulator controller to improve the error tracking accuracy of the manipulator controller when observing complex disturbances and noises. In addition, in order to solve the problem that the coupling between the external disturbances of each axis of the manipulator leads to a large amount of computation when observing disturbances, an improved full-order observer is designed, which simplifies the parameters of the controller combined with the BKS operator and reduces the complexity of the algorithm. Through a theoretical analysis and a simulation test, it was verified that the proposed manipulator controller could effectively suppress external disturbance and noise, and the application of the BKS operator in the manipulator servo system control is feasible and effective.

## 1. Introduction

The robotic manipulators have the characteristics of flexible movement and adaptability to harsh environments, and have been used in more and more fields in recent years, such as ocean exploration [1,2], health care [3], agricultural planting [4], and other fields. With the increase in manipulator application fields, its disturbance types are becoming more and more complex, so how to ensure its positioning accuracy and trajectory accuracy under complex disturbances has become an important research direction.

Aiming at the high-precision trajectory control of the manipulator, in recent years, many scholars have proposed a series of control and optimization methods to design and develop the controller of the manipulator. Shang et al. [5] proposed a PD controller with fuzzy logic combined with an RBF neural network optimization for the parameter perturbation caused by the telescopic flexible manipulator when the linkage length changes and the nonlinear rotation angle fluctuates, which reduces the trajectory error caused by the change of the manipulator structure parameters. Yang et al. [6] proposed an improved PID controller in order to identify the torque and stiffness of the manipulator, which improves the trajectory tracking accuracy of the manipulator by correcting the servo motor trajectory and satisfies the suppression of the disturbance generated by nonlinear friction. Fan et al. [7] designed a recursive neural network by collecting motion data of redundant manipulators and established a controller for the case of incomplete or unknown manipulator dynamics models to improve the trajectory tracking accuracy of industrial manipulators. The control method with an obvious effect on manipulator disturbance suppression is the sliding mode control, which can identify and track the disturbance for the characteristics of an external disturbance so that the accuracy of the manipulator’s controller trajectory tracking control can be improved [8,9]. It can be seen from the methods mentioned in the above documents that to improve the trajectory tracking accuracy and positioning accuracy of the manipulator, an accurate control model is required, and the controller needs to be improved to identify the noise and disturbance in the manipulator’s working environment. Therefore, designing an advanced observer for the controller is an effective method for improving the trajectory-tracking accuracy of the robotic manipulator.

By designing an appropriate disturbance observer to estimate the disturbance, and integrating the disturbance estimation into the manipulator controller, the negative effects of noise and disturbance on the trajectory tracking control can be greatly attenuated. Therefore, many scholars have studied and designed the disturbance observer. Shi et al. [10] have developed a new integral disturbance observer to actively compensate the uncertainty of the system and combined it with an adaptive sliding mode control to improve the transient performance of the manipulator controller. However, the simulation verification of this method on a 2-DOF robot manipulator cannot show the application effect on redundant robots that are widely used in industry. Gao and Chen [11] proposed a fixed-time extended state disturbance observer and proved its stability by improving the extended state observer. However, while this method can make the robot complete the task in a limited time, the global trajectory tracking accuracy cannot be guaranteed. Zhao et al. [12] proposed a boundary control based on the disturbance observer and explored that disturbance suppression can still be achieved under the premise that the time derivative of the disturbance received by the manipulator is not zero. However, this research uses a single-link flexible manipulator, which is difficult to apply to common manipulators. Jung and Lee [13] analyzed the application similarity of the disturbance observer and time delay controller in a manipulator, and found that the time delay controller can achieve better results in Cartesian space. However, the definition of manipulator disturbance is not complex enough to completely represent the case of manipulators in practical application. In recent years, some disturbance observers designed based on sliding mode observers have also achieved good results in the application of manipulator controllers, and the design of initial parameters is low in difficulty and easy to achieve. However, with the change of external disturbance characteristics, these parameters are difficult to evolve quickly and accurately, resulting in weak adaptability of the controller and difficulty applying in the presence of complex disturbances [14,15]. Therefore, a novel Bernstein–Kantorovich (BK) operator is added to the disturbance observer in this paper. By using its excellent approximation characteristics in statistics and data convergence estimation characteristics, the updating speed and accuracy of the parameters in the controller and disturbance observer are improved, and the identification accuracy of the manipulator controller for disturbance is improved.

The BK operator and its improved methods have been applied in the engineering field in recent years. They are mainly used to optimize the process of updating controller parameters to make it approach quickly and ensure accuracy [16]. Moreover, with the development of the BK operator, some improved BK operators, such as *λ*-BK [17,18], α-BK [19], q-BK [20], and other methods, have also been developed [21,22,23,24]. In this paper, a new BKS operator, which is introduced in [25], is implemented in the manipulator controller to optimize the parameter update process of the disturbance observer and improve the trajectory tracking accuracy of the robotic manipulator controller.

In this paper, we improve the robotic servo controller for the uncertainty of the robotic manipulator model and trajectory tracking problem due to the external unknown complex perturbations, and the main contributions of this paper are as follows:Combining the BKS operator with the perturbation observer in the robotic controller.A multi-turbulence simulation based on the new perturbation observer and the pre-researched trajectory tracking controller [26] on the robotic manipulator is performed to verify the effectiveness of the method.An experimental platform of the manipulator is built to verify the effectiveness of the proposed controller in the actual manipulator, which reflects the feasibility of the method in practical engineering applications.

## 2. Model Derivation and Control Objectives

Figure 1 shows the experimental setup. In this paper, a manipulator model is built, and motion control is realized by using a single-chip computer. The model is STM32F13RBT6. The trajectory optimization is updated and compensated by the upper computer. The insulation cup near the robot can be regarded as an obstacle or a coordinate reference. In this experimental environment, a vibration device is installed on the base to provide background noise; that is, the influence of noise, such as simulated mechanical vibration is added to the process of disturbance.

### 2.1. Manipulator Dynamic Modeling

The dynamic equation of an n-DOF manipulator under external disturbance can be given by the following equation:(1)$$M\left(q\right)\ddot{q}+C\left(q,\dot{q}\right)\dot{q}+G\left(q\right)+F\left(\dot{q}\right)=\tau +\tau_{d},$$
where $q,\;\dot{q},\;\mathrm{and}\;\ddot{q}\in R^{n}$ mean the vectors of manipulator joint positions, velocities, and accelerations, respectively. $M\left(q\right)\in R^{n\times n}$ is expressed as the positive definite inertia matrix of the manipulator joint; $C\left(q,\dot{q}\right)\in R^{n\times n}$ represents the Coriolis force and centrifugal force matrix of the manipulator joint in motion; $G\left(q\right)\in R^{n}$ represents the gravity matrix of the manipulator joint; $F\left(\dot{q}\right)\in R^{n}$ means the friction matrix, which is generally used to represent the dynamic friction force in the state of joint motion, and the static friction of starting is not considered here; $\tau \in R^{n}$ represents the control torque of each joint of the manipulator, which refers to the torque required by each joint of the manipulator to complete the task in an ideal state; $\tau_{d}\in R^{n}$ represents an uncertain external disturbance, which can also be regarded as the additional torque required by the manipulator to suppress the error caused by the external disturbance. In the application process of the actual robot manipulator, there are factors such as geometric errors, sudden load changes, and environmental noises, as well as uncertain emergencies such as obstacle avoidance, collisions, and human interference. These factors lead to the value of the physical parameters of the model. It cannot be obtained accurately, which will cause an error in the controller system parameters, and the uncertainty of the system can be expressed as:(2)$$\{\begin{array}{c} M\left(q\right)=M_{0}\left(q\right)-\Delta M\left(q\right)\\ C\left(q,\dot{q}\right)=C_{0}\left(q,\dot{q}\right)-\Delta C\left(q,\dot{q}\right)\\ G\left(q\right)=G_{0}\left(q\right)-\Delta G\left(q\right)\\ F\left(\dot{q}\right)=F_{0}\left(\dot{q}\right)-\Delta F\left(\dot{q}\right) \end{array},$$
where the symbol with the subscript 0, such as $M_{0}$, represents the nominal matrix of mechanical parameters of the robotic manipulator, and the symbol with Δ in front represents the parameter variation caused by uncertainty in the manipulator control system, such as ΔM.

Substituting Equation (2) into Equation (1), we can obtain the dynamic equation that can express the uncertainty of the manipulator.
(3)$$M\left(q\right)\ddot{q}+C\left(q,\dot{q}\right)\dot{q}+G\left(q\right)+F\left(\dot{q}\right)=\tau +\tau_{d}+\Delta M\left(q\right)\ddot{q}+\Delta C\left(q,\dot{q}\right)\dot{q}+\Delta G\left(q\right)+\Delta F\left(\dot{q}\right).$$

To organize the Equation (3) that expresses the system uncertainty, it can be written as:(4)$$\ddot{q}=M_{0}{\left(q\right)}^{-1}\left\{-C_{0}\left(q,\dot{q}\right)\dot{q}-G_{0}\left(q\right)-F_{0}\left(\dot{q}\right)+\tau +\tau_{d}+\Delta M\left(q\right)\ddot{q}+\Delta C\left(q,\dot{q}\right)\dot{q}+\Delta G\left(q\right)+\Delta F\left(\dot{q}\right)\right\}.$$

In order to facilitate subsequent calculation and derivation, the parameters in Equation (4) can be classified according to the characteristics of the dynamic model, which can be written as:(5)$$\ddot{q}=f_{0}\left(q,\dot{q}\right)+M_{0}{\left(q\right)}^{-1}\tau \left(t\right)+D\left(t\right),$$
where D(t) represents the error caused by system uncertainty and disturbance to the joint motion, which can also be regarded as the compensation of the manipulator to suppress the uncertainty. $f_{0}\left(q,\dot{q}\right)$ is a bounded and known nonlinear function obtained from the manipulator model.

In order to ensure the task trajectory accuracy of the robot manipulator, it is necessary to ensure its end joint motion state. By designing the sliding mode surface of the controller, the robustness of the sliding mode control is adopted to ensure that the controller can track the desired trajectory in the presence of system uncertainty.
(6)$$\underset{t\to t_{f}}{\lim}\left|e\right|=\underset{t\to t_{f}}{\lim}\left|q-q_{d}\right|=0,$$
where $q_{d}\in R^{n}$ represents the given task of the manipulator, that is, the expected trajectory; $e=q-q_{d}$ represents the trajectory tracking error caused by system uncertainty; $t_{f}$ is the expected time to complete the task, that is, the given finite time. The controller designed in this paper needs to establish the following assumptions and lemmas for the system and uncertainty to ensure the stability of the system.

It is assumed that the joint position and velocity state of the manipulator are data that can be measured by the joint or servo motor sensor. Uncertainty of ψD(t) meet $\left|\psi D\left(t\right)\right|\leq \omega_{0}$ and $\left|\psi \dot{D}\left(t\right)\right|\leq \omega_{1}$, where diagonal matrix ψ is a constant, which is based on mechanical properties of the manipulator. The constants $\omega_{0}$ and $\omega_{1}$ are unknown and positive.

When the above assumptions are satisfied, the stability of the following invariant systems can be proved according to the correlation lemma.
(7)$$\dot{z}=f\left(z\right).$$

Suppose there is a continuous function V(z) in the system shown in Equation (5) which satisfies the following conditions, i.e., it is a positive definite function on $D\subseteq R^{n}$ and satisfy the constraint condition:

If *V*(*z*) is a continuous function and $D\subseteq R^{n}$ is a positive definite function, the following constraints can be obtained:(8)$$\dot{V}\left(z\right)+kV^{\lambda }\left(z\right)\leq 0,$$When k>0 and λ∈(0, 1), it can be seen from Equation (8) that the defined manipulator control system is locally finite-time stable. In addition, if $D=R^{n}$ is satisfied, the defined system is globally finite-time stable.

It can be seen from the existing robot manipulator motion model that the constraint on the disturbance is mainly the amplitude of the disturbance, but the change characteristics of the disturbance are not discussed. From the actual application, it can be clearly seen that if the nonlinearity of the disturbance is too strong, the controller and observer have insufficient ability to suppress the disturbance, and the impact on the motion control accuracy is very obvious. To solve this problem, this paper proposes a modified control method of a robotic manipulator which can ensure high-precision motion control when the manipulator works in complex environments such as underwater, picking, and so on.

### 2.2. Bernstein-Kantorovich-Stancu Operator

In this section, we apply the BKS operator to the manipulator and obtain the recursive relationship of BKS polynomial according to the Bernstein polynomial [27]. Then, some properties of the BKS operator applied to the robotic manipulator controller are given according to the above-derived model.

The modified BKS operator is defined in [25] as
(9)$$K_{n,\lambda_{i}}^{\left(\alpha ,\beta \right)}\left(f;x\right)=\left(n+\beta +1\right)\sum_{i=0}^{n}{\tilde{b}}_{i,n}\left(x\right)\int_{\frac{i+\alpha }{n+\beta +1}}^{\frac{i+\alpha +1}{n+\beta +1}}f\left(t\right)dt,x\in \left[0,1\right],$$
where 0≤α≤β and f represent a continuous function defined on the interval [0,1].

It is clear that each of these nth BKS basis polynomials are a linear combination of Bernstein basis polynomials for the higher order observer coefficients of complex manipulators, and the higher order combination of BKS basis polynomials is proved below.

The Bernstein basis polynomial [25] can be defined as $b_{i,n}\left(x\right)=C_{n}^{i}x^{i}{\left(1-x\right)}^{n-i}$;
(10)$${\tilde{b}}_{i,n}\left(x\right)=\{\begin{array}{ll} \frac{C_{n-1}^{i}+\lambda_{i-1}}{C_{n}^{i}}b_{i,n}\left(x\right)+\frac{C_{n-1}^{i}-\lambda_{i}}{C_{n}^{i+1}}b_{i+1,n}\left(x\right) & ;i=0,1,\ldots ,\lfloor \frac{n}{2}\rfloor -1\\ \frac{2\lambda_{i-1}}{C_{n}^{i}}b_{i,n}\left(x\right) & ;i=\lfloor \frac{n}{2}\rfloor \\ \frac{C_{n-1}^{i-1}-\lambda_{i-2}}{C_{n}^{i-1}}b_{i-1,n}\left(x\right)+\frac{C_{n-1}^{i-1}+\lambda_{i-1}}{C_{n}^{i}}b_{i,n}\left(x\right) & ;i=\lfloor \frac{n}{2}\rfloor +1,\ldots ,n \end{array},$$

For the sliding mode observer, which has an order up to the second order, it is necessary to derive the case when n is an even number.
(11)$${\tilde{b}}_{i,n}\left(x\right)=\{\begin{array}{ll} \frac{C_{n-1}^{\left\lfloor \frac{n}{2}\right\rfloor }\lambda_{i-1}^{n}}{C_{n}^{i}}b_{i,n}\left(x\right)+\frac{C_{n-1}^{i}-\lambda_{i}}{C_{n}^{i+1}}b_{i+1,n}\left(x\right) & ;i=0,1,\ldots ,\lfloor \frac{n}{2}\rfloor -1\\ \frac{C_{n-1}^{i}+\lambda_{i-1}}{C_{n}^{i}}b_{i,n}\left(x\right) & ;i=\lfloor \frac{n}{2}\rfloor ,\lfloor \frac{n+1}{2}\rfloor \\ \frac{C_{n-1}^{i-1}-\lambda_{i-2}}{C_{n}^{i-1}}b_{i-1,n}\left(x\right)+\frac{C_{n-1}^{i-1}+\lambda_{i-1}}{C_{n}^{i}}b_{i,n}\left(x\right) & ;i=\lfloor \frac{n+1}{2}\rfloor +1,\ldots ,n \end{array},$$
where ⌊∗⌋ denotes the floor function, and $C_{n}^{i}=\left(\begin{array}{c} n\\ i \end{array}\right)$ for n≥i≥0. The associated shape parameters $\lambda_{i}$ are constrained to be $\lambda_{-1}=\lambda_{n-1}=0$ [25].

However, if the observer is of full order, and n is odd, then it is called an nth extended Bernstein basis polynomial and the associated parameters satisfy the following conditions:(12)$$\lambda_{i},\lambda_{n-2-i}\in \left[-C_{n-1}^{i+1},C_{n-1}^{i}\right],i=0,1,\ldots ,\lfloor \frac{n-1}{2}\rfloor -1.$$

By derivation, it follows that in case *λ_i_*, α, and β are any real numbers satisfying the above conditions, the following moments were obtained in [25]
(13)$$\begin{array}{l} K_{n,\lambda_{i}}^{\left(\alpha ,\beta \right)}\left(e_{0};x\right)=1\\ K_{n,\lambda_{i}}^{\left(\alpha ,\beta \right)}\left(e_{1};x\right)=\frac{n}{\left(n+\beta +1\right)}x+\frac{2\alpha +1}{2\left(n+\beta +1\right)}+D_{n,\lambda_{i},1}^{\alpha ,\beta }\left(x\right)\\ K_{n,\lambda_{i}}^{\left(\alpha ,\beta \right)}\left(e_{2};x\right)=\frac{n\left(n-1\right)}{{\left(n+\beta +1\right)}^{2}}x^{2}+\frac{2n\left(\alpha +1\right)}{{\left(n+\beta +1\right)}^{2}}x+\frac{3\alpha^{2}+3\alpha +1}{3{\left(n+\beta +1\right)}^{2}}+D_{n,\lambda_{i},2}^{\alpha ,\beta }\left(x\right)\\ K_{n,\lambda_{i}}^{\left(\alpha ,\beta \right)}\left(e_{3};x\right)=\frac{n\left(n-1\right)\left(n-2\right)}{{\left(n+\beta +1\right)}^{3}}x^{3}+\frac{3n\left(n-1\right)\left(2\alpha +3\right)}{2{\left(n+\beta +1\right)}^{3}}x^{2}+\frac{6\alpha^{2}n+12\alpha n+7n}{2{\left(n+\beta +1\right)}^{3}}x\\ \;+\frac{4\alpha^{3}+6\alpha^{2}+4\alpha +1}{4{\left(n+\beta +1\right)}^{3}}+D_{n,\lambda_{i},3}^{\alpha ,\beta }(x) \end{array}$$
where
(14)$$\begin{array}{c} D_{n,\lambda_{i,k},k}^{\alpha ,\beta }\left(x\right)=\frac{1}{\left(k+1\right){\left(n+\beta +1\right)}^{k}}\overset{\left\lfloor \frac{n}{2}\right\rfloor }{\underset{i=1}{\sum }}\left(\Delta_{i-1,k}^{\alpha }-\Delta_{i,k}^{\alpha }\right)\left[\frac{C_{n-1}^{i-1}-\lambda_{i-1}}{C_{n}^{i}}\right]b_{i,n}\left(x\right)\\ +\frac{1}{\left(k+1\right){\left(n+\beta +1\right)}^{k}}\overset{n-1}{\underset{i=\lfloor \frac{n}{2}\rfloor }{\sum }}\left(\Delta_{i+1,k}^{\alpha }-\Delta_{i,k}^{\alpha }\right)\left[\frac{C_{n-1}^{i-1}-\lambda_{i-1}}{C_{n}^{i}}\right]b_{i,n}\left(x\right). \end{array}$$

Based on the form of Equation (13), according to the role of dynamics in the movement process of the robot arm, we can use it in the dynamic model of the robot arm in two ways. One is that the time when the sliding mode controller passes through the sliding mode surface is taken as the starting point of the sampling time, and the subsequent time when the sliding mode controller continuously passes through the sliding mode surface is taken as the subsequent sampling time. The other way is to set the boundary layer of the sliding mode control. When the control function output crosses the boundary layer, it is used as the starting point of the sampling time. If the output is always outside the preset boundary layer, the sampling time interval is set according to the actual application results. Both methods have corresponding application scenarios. The former is suitable for the continuous occurrence of external force disturbance, while the latter is suitable for the occurrence of disturbance according to impulse noise.

In order to ensure that the operator converges in the course of the iteration, it is necessary to prove $D_{n,\lambda_{i,k},k}^{\alpha ,\beta }\left(x\right)$. The range of its interval can be obtained as follows:(15)$$\begin{array}{l} D_{n,\lambda_{i},k}^{\alpha ,\beta }\left(x\right)=\frac{1}{\left(k+1\right){\left(n+\beta +1\right)}^{k}}\overset{\left\lfloor \frac{n}{2}\right\rfloor }{\underset{i=1}{\sum }}\left(\Delta_{i-1,k}^{\alpha }-\Delta_{i,k}^{\alpha }\right)\left[\frac{C_{n-1}^{i-1}-\lambda_{i-1}}{C_{n}^{i}}\right]b_{i,n}\left(x\right)+\\ +\frac{1}{\left(k+1\right){\left(n+\beta +1\right)}^{k}}\overset{n-1}{\underset{i=\lfloor \frac{n}{2}\rfloor }{\sum }}\left(\Delta_{i+1,k}^{\alpha }-\Delta_{i,k}^{\alpha }\right)\left[\frac{C_{n-1}^{i}-\lambda_{i-1}}{C_{n}^{i}}\right]b_{i,n}\left(x\right)\\ \leq \frac{1}{\left(k+1\right){\left(n+\beta +1\right)}^{k}}\overset{n}{\underset{i=0}{\sum }}\left(\Delta_{i+1,k}^{\alpha }-2\Delta_{i,k}^{\alpha }+\Delta_{i-1,k}^{\alpha }\right)b_{i,n}\left(x\right)\\ \leq \frac{{\left(n+\alpha +2\right)}^{k+1}-3{\left(n+\alpha +1\right)}^{k+1}+3{\left(n+\alpha \right)}^{k+1}-{\left(n+\alpha -1\right)}^{k+1}}{\left(k+1\right){\left(n+\beta +1\right)}^{k}}. \end{array}$$

In addition, the following results were obtained in [25]:(16)$$\underset{n\to \infty }{lim}-\frac{6C_{k+1}^{3}n^{k-2}}{{\left(n+\beta +1\right)}^{k}}\leq \underset{n\to \infty }{lim}D_{n,\lambda_{i},k}^{\alpha ,\beta }\left(x\right)\leq \underset{n\to \infty }{lim}\frac{6C_{k+1}^{3}n^{k-2}}{{\left(n+\beta +1\right)}^{k}}.$$
and
(17)$$\underset{n\to \infty }{lim}D_{n,\lambda_{i},k}^{\alpha ,\beta }\left(x\right)=0.$$

Therefore, it can be proved that the modified BKS operator proposed in this paper does not cause the divergence of the control system due to the change of the operator coefficients caused by the perturbation when the alternative parameters are involved in the calculation in the control system and can be applied in the manipulator control system.

## 3. Controller Design

The design of the manipulator controller needs to be based on the actual engineering task requirements. In order to more clearly express the dynamic control of the manipulator in the process of completing the task, a simple definition of the finite-time generator is created. It is mainly aimed at the dynamic control of manipulator tasks, assuming that there is a double integral system $\ddot{e}\left(t\right)=u\left(t\right)$ in the manipulator. According to the work in [28], in order to achieve local convergence of the manipulator control system within the desired target time of the task, it is necessary to ensure that the input u(t) in the controller tracks the desired trajectory of e(t) and $\dot{e}\left(t\right)$.

### 3.1. Manipulator’s Task Definition

This section focuses on the manipulator configuration error caused by the difference between the given motion trajectory output by the controller and the expected trajectory of the human-made task when the robot manipulator performs the task, and the mapping relationship between the coordinates in the manipulator system and the actual motion coordinates. The error function can be written as:(18)$$e=x\left(q\right)-x^{d}.$$

We need to assume that Equation (18) has a quadratic differential with respect to time, and then define that x(q) is the trajectory of the manipulator obtained from the solution of the positive kinematics, and $x^{d}$ is the desired trajectory defined in the given task requirements, then $\dot{e}=\dot{x}=J\dot{q}$ where J=∂e/∂q is the Jacobian matrix expression of the desired trajectory of the task.
(19)$$\ddot{e}=J\ddot{q}+\dot{J}\dot{q}.$$

Substitute Equation (5) into (19) to obtain the following form:(20)$$\ddot{e}=Q\tau +\mu ,$$
where $Q=JA^{-1}$ and $\mu =-Qb+\dot{J}\dot{q}$ are the influence of the error between the expected trajectory and the actual trajectory of the manipulator during a given task on the motion control of the manipulator. Then we can obtain the dynamic model of the manipulator movement under the influence of the tasks as follows:(21)$$\tau =Q^{A}\left(u-\mu \right),$$
where
(22)$$Q^{A}=AQ^{T}[QAQ^{T}{]}^{-1}=J^{T}[JA^{-1}J^{T}{]}^{-1}$$
means the generalized matrix inverse of $JA^{-1}$, and u represents the parameter used to adjust the input of the manipulator controller.

In defining the manipulator dynamics model, it is mentioned that the inertia matrix, A, is a positive definite matrix, so the following assumptions can be given:(23)$$A_{min}\leq \left|\left|A^{-1}\right|\right|\leq A_{max}<\infty ,$$
where $A_{min}$ and $A_{max}$ are defined as positive constants.

In addition, we also need to define another constant α to satisfy the constraints of 0<α<1 and
(24)$$\left|\left|A^{-1}\tilde{A}-I\right|\right|\leq \alpha ,$$
where the uncertainty of the dynamic model of the manipulator controller in the above model derivation section is represented by $\tilde{A}$.

Based on this, the manipulator dynamics model can be written as
(25)$$M=JA^{-1}J^{T}{\left[J{\tilde{A}}^{-1}J^{T}\right]}^{-1}=QQ^{A};\;\left|\left|M\right|\right|=\left|\left|QQ^{A}\right|\right|\leq M_{max}$$
where $M_{max}$ is a normal number representing the upper limit of the inertia matrix variation. It is worth mentioning that if there is no other interference, that is, satisfying $A=\tilde{A}$, then M=I can be obtained.

### 3.2. The Designed Controller

To solve the trajectory tracking problem using only the position feedback, the control inputs and observer dynamics can be written as:(26)$$u\left(t\right)=-K_{d}\dot{\hat{e}}-K_{p}\hat{e}+{\ddot{q}}_{d}-\hat{D}\left(t\right),$$
(27)$$\{\begin{array}{l} \dot{\hat{e}}=\rho +L_{d}\left(e-\hat{e}\right)\\ \dot{\rho }=L_{p}\left(e-\hat{e}\right) \end{array},$$
where $K_{d}$ and $K_{p}$ are positive the diagonal gain matrices; $\hat{e}\left(t\right)=\hat{q}\left(t\right)-q_{d}\left(t\right)$ is the approximate accurate value observer output of e(t); $L_{d}$ and $L_{p}$ are the observer gain matrices, and $\hat{D}\left(t\right)$ represents the estimation of D(t). The following assumptions are considered for the values of the above parameters:(28)$$\begin{array}{c} K_{p}=\mu_{d}K_{d}\\ K_{d}=\left(k_{d}+\gamma \right)I_{n}\\ L_{p}=\mu_{d}l_{d}I_{n}\\ L_{d}=\left(l_{d}+\mu_{d}\right)I_{n} \end{array},$$
where $\mu_{d}$, $k_{d}$, *γ* and $l_{d}$ are positive constants, satisfying $k_{d}>\mu_{d}$, and $I_{n}$ shows the identity matrix of dimension n. Substituting Equation (26) into Equation (20) and using the definition of Equation (28) yields
(29)$$\ddot{e}=-K_{d}\dot{\hat{e}}-\mu_{d}K_{d}\hat{e}+D\left(t\right)-\hat{D}\left(t\right).$$

Then we define
(30)$$s_{1}\left(t\right)=\dot{e}\left(t\right)+\mu_{d}e\left(t\right),$$
(31)$$s_{2}\left(t\right)=\dot{q}\left(t\right)-{\dot{q}}_{0}\left(t\right).$$
where ${\dot{q}}_{0}\left(t\right)=\dot{\hat{q}}\left(t\right)-\mu_{d}\tilde{q}\left(t\right)$ and $\tilde{q}\left(t\right)=e\left(t\right)-\hat{e}\left(t\right)=q\left(t\right)-\hat{q}\left(t\right)$ denotes the position estimation error of the manipulator joints. Therefore, we can write the closed-loop error dynamics equation of the system as:(32)$$\ddot{e}=-K_{d}s_{1}+K_{d}s_{2}+D\left(t\right)-\hat{D}\left(t\right)$$
(33)$${\dot{s}}_{2}+\left(l_{d}I_{n}-K_{d}\right)s_{2}=-K_{d}s_{1}+D\left(t\right)-\hat{D}\left(t\right),$$

In Section 2.2 we proved that the proposed BKS operator is a mathematical tool, which is a simplified calculation and it is easier for making polynomials used to estimate different functions. By combining multiple polynomials, the BKS operator is able to estimate any complex and real-valued function with any degree of accuracy. The ideal form of the dynamic equation of the robotic manipulator can be regarded as a linear system. The traditionally obtained algebraic linear system is derived from the least squares solution of the minimum norm using accurate data. However, the introduced disturbance cannot be completely regarded as a nonlinear part. Some complex noise disturbances are difficult to be completely distinguished from the linear system in the discrete sampling process. Therefore, we need to adopt a method to suppress this noise as much as possible while retaining the linear system, even if the basis function of the control system is smoother. The BKS can introduce a regularization feature to obtain a more smooth solution. In Equation (26), u(t) is the auxiliary function for constructing the sliding mode controller and the dynamic model in Equation (5) is substituted. However, the mean square error of u(t) is approximated to the unknown function in the first kind of integral equation by the given form of the BKS operator. The obtained linear equations are transformed into algebraic linear equations, and a more stable numerical solution can be obtained after approximating with a higher order modified BKS operator. According to the design of the controller, we rewrite Equation (9) as follows:(34)$$K_{mp}^{\left(\alpha ,\beta ,\gamma \right)}\left(f,t\right)=\sum_{k=0}^{m+p}\left(C_{k}^{m+p}\right)\frac{t^{\left(k,-\alpha \right)}{\left(1-t\right)}^{\left(m+p-k,-\alpha \right)}}{1^{\left(m+p,-\alpha \right)}}f\left(\frac{k+\beta }{m+\gamma }\right).$$

In Equation (34), $C_{k}^{m+p}$ denotes the binomial coefficient, which is described as
(35)$$C_{k}^{m+p}=\frac{\left(m+p\right)!}{\left(m+p-k\right)!k!}.$$

The kth order factorial power with increment (−α) of t is shown by $t^{\left(k,-\alpha \right)}$, which is described as
(36)$$t^{\left(k,-\alpha \right)}=\{\begin{array}{cc} \prod_{v=0}^{k-1}\left(t+v\alpha \right) & \mathrm{if}\;k\geq 1\;\mathrm{and}\;t>0\\ 1 & \mathrm{else} \end{array}.$$

For ease of representation, it can be directly presented as:(37)$$K_{mp}^{\left(\alpha ,\beta ,\gamma \right)}\left(f,t\right)=W_{f}^{T}Z_{f},$$
where
(38)$$W_{f}={\left[f\left(\frac{\beta }{m+\gamma }\right)\left(m+p\right)f\left(\frac{1+\beta }{m+\gamma }\right)\frac{\left(m+p\right)\left(m+p-1\right)}{2!}f\left(\frac{2+\beta }{m+\gamma }\right)\;\ldots \;f\left(\frac{m+p+\beta }{m+\gamma }\right)\right]}^{T},$$
and
(39)$$Z_{f}={\left[\frac{{\left(1-t\right)}^{\left(m+p,-\alpha \right)}}{1^{\left(m+p,-\alpha \right)}}\frac{t^{\left(1,-\alpha \right)}{\left(1-t\right)}^{\left(m+p-1,-\alpha \right)}}{1^{\left(m+p,-\alpha \right)}}\frac{t^{\left(2,-\alpha \right)}{\left(1-t\right)}^{\left(m+p-2,-\alpha \right)}}{1^{\left(m+p,-\alpha \right)}}\;\ldots \;\frac{t^{\left(m+p,-\alpha \right)}}{1^{\left(m+p,-\alpha \right)}}\right]}^{T}.$$

Therefore, we can express the basis function for D(t) using a linear form based on the BKS and approximation theorem as
(40)$$D\left(t\right)=W_{D}^{T}Z_{D}+\varepsilon_{D},$$
where the number of basis functions used is determined by the filtering accuracy of the controller, and the ideal weighting vector is given by $W_{D}$. The approximation error denoted by $\varepsilon_{D}$ and $Z_{D}$ is the basis function vector. The $W_{D}$, $Z_{D}$, and $\varepsilon_{D}$ are determined by the filter and observer in the controller. The selection of the basis function parameters has an impact on the filtering accuracy. By establishing the relationship between the probability density function of the filter estimation error and the filter gain matrix, the filtering dynamic system is uniformly bounded in the mean square sense, indicating that the feedback particle filter will cause system divergence at low sampling rates. In this paper, full order observers are used to determine the filtering accuracy. Using the same set of basis functions, it can be written as
(41)$$\hat{D}\left(t\right)={\tilde{W}}_{D}^{T}Z_{D}.$$

Next, substituting Equation (40) and Equation (41) into Equation (32) and Equation (33), we can conclude that
(42)$$\ddot{e}=-K_{d}s_{1}+K_{d}s_{2}+{\tilde{W}}_{D}^{T}Z_{D}+\varepsilon_{D};$$
(43)$${\dot{s}}_{2}+\left(l_{d}I_{n}-K_{d}\right)s_{2}=-K_{d}s_{1}+{\tilde{W}}_{D}^{T}Z_{D}+\varepsilon_{D}$$
where the BKS operator weight approximation error is denoted by ${\tilde{W}}_{D}$.

A simulation case was used to verify the performance of the proposed manipulator controller combined with the BKS operator. The trajectory of each joint used in the simulation, the controller inputs, and the perturbations added to the joint #1 are shown in Figure 2. The simulation results are shown in Figure 3. During the activation of the joint servo motor, a disturbance exists between 6 and 8 s. Figure 3a shows the controller without the BKS operator, which lacks sufficient suppression capability in the face of the externally imposed disturbance, while the controller with the BKS operator applied has a higher accuracy in tracking the disturbance, as shown in Figure 3b, where the impact caused by the imposed disturbance is significantly reduced.

However, it is worth noting that only one joint of the manipulator is perturbed in this simulation. Due to the series connection of the manipulator, the disturbance of one joint will affect the other joints. Taking the above simulation as an example, if all three joints are disturbed, then there is a complex coupling relationship between the total disturbance of each joint. When the controller uses the standard model to calculate such problems, it will consume a lot of time, which cannot meet the requirements of real-time control in the actual application of the manipulator and will have a great impact on the communication throughput. We propose an effective solution to this problem.

### 3.3. Design of the Modified Observer

Real-time control is an important parameter index of the manipulator, so the observer in the articulated rotary axis servo control system should consider more influencing factors at the same time, including parameter observation accuracy, observation speed, parameter setting, and computational complexity [29]. The controller method proposed above can only guarantee the observation accuracy in the servo control system, so a new full-order state observation scheme for manipulator servo systems is designed. Combining the BKS operator allows the controller to take into account both observation performance and execution capability. Combining the BKS operator allows the controller to achieve a compromise between system tracking capability and noise sensitivity by introducing only one adjustable parameter. At the same time, determining whether the value of the parameter is reasonable can optimize the performance of the required observer, thereby simplifying the controller parameter adjustment steps.

Considering that there is a transmission device such as a reducer in the servo motor of the manipulator, the observer of the manipulator servo system actually considers the data of two rotating bodies. So, the models of two observers can be designed at the same time, that is, the motor side of the joint and the actuator. The state directly observed on the motor side includes the motor position ${\hat{\theta }}_{m}$ and speed ${\hat{\omega }}_{m}$. From this, the motor-side observer model can be obtained:(44)$$\{\begin{array}{c} {\tilde{\theta }}_{m}\left(k\right)={\hat{\theta }}_{m}\left(k-1\right)+T{\hat{\omega }}_{m}\left(k-1\right)\\ +\frac{T^{2}}{2}\left[\frac{T_{e}\left(k-1\right)-{\hat{T}}_{s}\left(k-1\right)-b_{1}{\hat{\omega }}_{m}\left(k-1\right)}{J_{1}}\right]\\ {\tilde{\omega }}_{m}\left(k\right)={\hat{\omega }}_{m}\left(k-1\right)\\ +T\left[\frac{T_{e}\left(k-1\right)-{\hat{T}}_{s}\left(k-1\right)-b_{1}{\hat{\omega }}_{m}\left(k-1\right)}{J_{1}}\right]\\ {\tilde{a}}_{m}\left(k\right)={\hat{a}}_{m}\left(k-1\right). \end{array}.$$

In the designed motor-side observer, the motor position $\theta_{m}$ measured by the encoder can be used to correct the estimated value in real time, so as to meet the requirement of the observer in order to continuously improve the observation accuracy. We use the difference Δθ(k) between the directly measured actual value and the predicted estimated value as one of the input signals for the next sampling point of the motor-side observer. In addition, the driver of the servo motor can also obtain the electromagnetic torque $T_{e}$ in real time by measuring the q-axis current. Thus, the transfer torque $T_{s}$ can be obtained indirectly by observing the motor speed and acceleration derived from Equation (45).
(45)$${\hat{T}}_{s}\left(k\right)=T_{e}\left(k\right)-b_{1}{\hat{\omega }}_{m}\left(k\right)-J_{1}{\hat{a}}_{m}\left(k\right).$$

Directly observed conditions, include load position ${\hat{\theta }}_{l}$, speed ${\hat{\omega }}_{l}$, and acceleration ${\hat{a}}_{l}$.
(46)$$\{\begin{array}{c} {\tilde{\theta }}_{l}\left(k\right)={\hat{\theta }}_{l}\left(k-1\right)+T{\hat{\omega }}_{l}\left(k-1\right)\\ +\frac{T^{2}}{2}\left[\frac{T_{s}\left(k-1\right)-{\hat{T}}_{L}\left(k-1\right)-b_{2}{\hat{\omega }}_{l}\left(k-1\right)}{J_{2}}\right]\\ {\tilde{\omega }}_{l}\left(k\right)={\hat{\omega }}_{l}\left(k-1\right)\\ +T\left[\frac{T_{s}\left(k-1\right)-{\hat{T}}_{L}\left(k-1\right)-b_{2}{\hat{\omega }}_{l}\left(k-1\right)}{J_{2}}\right]\\ {\tilde{a}}_{l}\left(k\right)={\hat{a}}_{l}\left(k-1\right). \end{array}.$$

By observing the load speed and acceleration, the load torque $T_{L}$ can be indirectly observed. The equation is as follows:(47)$${\hat{T}}_{L}\left(k\right)={\hat{T}}_{s}\left(k\right)-b_{2}{\hat{\omega }}_{l}\left(k\right)-J_{2}{\hat{a}}_{l}\left(k\right).$$

Considering that the load-side observer does not have a direct input signal, it is necessary to obtain the position difference between the motor-side and the load-side and solve the following differential equation:(48)$$\frac{d\left(\theta_{m}-\theta_{l}\right)}{dt}+\frac{K_{s}}{c_{s}}\left(\theta_{m}-\theta_{l}\right)=\frac{{\hat{T}}_{s}}{c_{s}}.$$

In Equation (48), $T_{s}$ is the value obtained by the observer on the motor-side, and $\theta_{m}$ is measured by the encoder of the motor. The purpose of correcting the state observation is achieved by solving the equation, and then the input signal of the load-side observer is constructed.

The parameter form of the BKS operator in the fixed gain matrix of the controller can be further derived as follows:(49)$$K={\left[\begin{array}{ccc} \alpha & \frac{\beta }{T} & \frac{2\gamma }{T^{2}} \end{array}\right]}^{T},$$
where α, β, and γ are dimensionless constants that can actually be solved analytically, and the solution can be represented by a parameter, that is, the ratio of motion state to observation uncertainty, defined as:(50)$$\lambda =\frac{T^{2}\sigma_{w}}{\sigma_{v}}.$$

Then, all feedback gains from the servo system observer can be derived in terms of λ. When this parameter is obtained, the optimal steady-state gain parameters α, β, and γ, as well as the resulting performance, can be obtained by Equation (51)
(51)$$\{\begin{array}{c} \lambda =\frac{2\gamma }{\sqrt{1-\alpha }}\\ \begin{array}{c} \gamma =\frac{\beta^{2}}{4\alpha }\\ \beta =4-4\sqrt{1-\alpha }-2\alpha \end{array} \end{array}.$$

The influence of traditional observer parameters on observation performance is not obvious. In order to simplify the process of parameter formulation and to keep the stability of the best gain matrix of the observer, an adjustable parameter $s=\sqrt{1-\alpha }$ is introduced. At this point, through the explicit expression of the elements in K, s can be obtained, as shown in Equation (52). By analyzing the stability of the observer, the value range of s can be refined.
(52)$$\{\begin{array}{c} \alpha =1-s^{2}\\ \beta =2{\left(s-1\right)}^{2}\\ \gamma ={\left(1-s\right)}^{3}/\left(s+1\right) \end{array}.$$

The relationship between s and λ can then be derived as shown in Equation (53). By analysis, s is a monotonically decreasing function of λ. Since λ > 0 and s∈(0,1), the performance of the observer can be adjusted according to s.
(53)$$s=\frac{\sigma }{6}-\frac{\lambda }{6}+\frac{\lambda \left(\lambda -18\right)}{6\eta }+1,$$
where $\eta =\sqrt[{3}]{27\lambda^{2}-108\lambda -\lambda^{3}+3\sqrt{3}\lambda \sqrt{432-\lambda^{2}}}$.

In order to ensure the stability of the controller of the manipulator, it is necessary to refer to the poles and zeros of the system to realize the requirement of adjusting the amplitude and phase of the unit impulse response of the system. Therefore, the value interval of s can be determined by the position of the pole in the characteristic equation of the observer. From the perspective of the servo system model of the manipulator joints, the characteristic equation of the relationship between the state matrix and the system output can be expressed as:(54)$$ℜ\left(z\right)=\left(-s-1\right)z^{3}+\left(7s-1\right)z^{2}+\left(s^{3}-7s^{2}\right)z+\left(s^{3}+s^{3}\right).$$

According to Equation (54), the transfer function between the input signal and the observed state shows that the amplitude of Equation (55) is equal to 1 when z=1. This indicates that after reaching steady state, the observation of the position is able to track the actual value without error, regardless of the input signal.
(55)$$G_{\theta }\left(z\right)=\frac{\hat{\theta }\left(k\right)}{\theta \left(k\right)}=\frac{\left(s^{3}+s^{2}-s-1\right)z^{3}+\left(-3s^{3}-3s^{2}+7s-1\right)z^{2}+4\left(s^{3}-s^{2}\right)z}{ℜ\left(z\right)}$$

## 4. Simulation and Discussion

This section will first explain the calculation results of the BKS operator in the process of manipulator controller parameter tuning and give appropriate controller parameters. Then, four disturbance cases are used to test the manipulator model to verify that the proposed controller can suppress the disturbance in the manipulator control process. Finally, to illustrate the role of the proposed observer in the controller, signals with a certain mechanical vibration noise are used to show the tracking effect of the proposed observer on noise and disturbance.

In the simulation, we use a CPU with Intel Core i7-7700@4.20. A personal computer with GHz and 8 GB RAM uses MATLAB 2021a to simulate the designed controller.

### 4.1. Disturbance Cases and Control Parameters

In the subsequent simulation, we introduced the following four kinds of disturbance properties into the three joints of the manipulator: two cases with different joint strengths between 6–8 s and two cases with multiple high-frequency disturbances between 5–9 s. The forms of the four disturbances are shown in Figure 4.

In the simulation, the degree of freedom of the manipulator is 6, so we can take n=6. At this time, the error between the trajectory obtained by using the basis function of $\hat{D}\left(t\right)$ determined by the BKS operator in the numerical simulation and the expected trajectory is shown in Table 1. The comparison with other BK operators shows the advantages of the BKS operator proposed in this paper.

As shown in Table 1, compared with the classical BK operator [18], q-BK [24] (q=0.6), α-BK [19,30] (α=0.5), and λ-BK [31] (λ=−0.3) make a comparison, and the error of the BKS operator proposed in this paper is smaller. Under the condition that n=6 of the model in this paper is satisfied, the error is the smallest. If we conduct iterative calculation, when n=40 is tested, it can obtain a more reasonable basis function, which makes the error of the model lower.

We can adjust the value of s according to the different needs of the user for the manipulator and the adaptation to the environment. For example, if the observer is used for a task that requires a high completion time, the observation speed is the most important indicator, so the s should be designed to be appropriately decreased. If it is a high-precision system, such as precision machining, or a system that pursues high data smoothness, the value of A should be appropriately increased. In addition, adjustable parameters can be adjusted according to the different observation requirements. In addition, the adjustable parameters can be optimized under different observation requirements. Considering the requirements of most working conditions, s=0.55 is selected as the parameter of the observer in this paper. The parameters of the BKS operator are α=0.5, β=0.7, and γ=0.1. The parameters of the controller are shown in Table 2.

### 4.2. Simulation Results of the Proposed Controller Using the BKS Operator

In this section, the four disturbance cases shown in Figure 4 are used to verify the controller’s ability to suppress different disturbances in different disturbance environments by taking the starting process of the servo motors at each joint of the manipulator as the experimental environment. The control effects of the improved means proposed in this paper on various characteristic disturbances are illustrated, respectively, by using and not using the BKS operator controller proposed in this paper.

It can be seen from Case 1 and Case 2 in Figure 4 that the difference between the first and the second disturbance is that the amplitude of the disturbance applied to the three joints is different, but it can be seen from Figure 5 and Figure 6 that the vibration caused by the disturbance with a large amplitude applied in Case 2 during the starting process of the servo motor is less obvious than that in Case 1 because there is a coupling relationship between the disturbances between joints. That is to say, the disturbance result of the front joint may be applied to the joint at the end, and the disturbance compensation of the end joint will affect the control signal of the front joint. Therefore, it is not possible to judge whether the stability of the manipulator servo system is severely affected by the given disturbance amplitude strength alone, but when it is compared through Figure 5a, Figure 5b, Figure 6a, and Figure 6b, respectively, it can be seen that the controller with the BKS operator proposed in this paper can effectively suppress the influence of external disturbance on the manipulator servo motor.

Figure 7 and Figure 8, respectively, show the simulation results of Case 3 and Case 4 shown in Figure 4. The disturbances in these two cases are high-frequency and have a longer duration. It can be seen from the simulation results that in this case, if the control method of the BKS operator proposed in this paper is not applied, the applied disturbance cannot be suppressed. Consequently, the controller cannot effectively suppress the disturbance during the application process, which shows that the manipulator servo control system with the BKS operator proposed in this paper can effectively suppress the complex disturbance, and it is effective and feasible.

In addition, there is another thing to note. From the comparison between (a) and (b) in Figure 5, Figure 6, Figure 7 and Figure 8, it can be seen that the starting stability time of the servo system without the BKS operator proposed in this paper is longer than that of the controller with the BKS operator, which shows that the controller with the BKS operator proposed in this paper can not only improve the stability of the manipulator in complex disturbance environments but also improve the starting performance of the servo system.

### 4.3. Experiment Results of the Manipulator Controller in a Complex Environment

In order to verify the advantages of the robotic manipulator control method proposed in this paper, we compared it with the new method in the same field in [14] and used the disturbance of Case 2 in Figure 4 carried out under the condition that the disturbance has no background noise and there is background noise in the disturbance–to illustrate the effectiveness of the method in this paper.

In this part, we used a manipulator model with an open source driver and compared its application effect with other methods in the model. This also reflects the positive influence of the servo control method proposed in this paper on the trajectory tracking control of the controller. The control flow of the model is shown in Figure 9.

As shown in the position estimation and velocity estimation in Figure 10, the method in [14] can observe the position and velocity during the operation of the manipulator servo system, and as shown in the disturbance estimation in Figure 10, it can better control the manipulator joints. However, in the initial stage of the servo system operation, that is, during the start-up process, it can be seen that there are obvious observation distortions. As shown by the error between the estimated result and the given result, the method is stable when observing the position, speed, and disturbance. There are obvious errors, especially the observation error, which is very large in the starting process. Compared with the disturbance imposed in Figure 6, Figure 7 and Figure 8, the influence on the stability of the system is not obvious. This shows that the method proposed in [14] is not timely enough to update the model after the system state changes, and it is not suitable for manipulators that require high-precision positioning accuracy and trajectory accuracy.

The experimental results of the method proposed in this paper are shown in Figure 11. Compared with Figure 10, both the position estimation accuracy and the speed estimation accuracy are improved. Especially for the tracking of the disturbance error, not only is the estimation error of the disturbances between Figure 6, Figure 7 and Figure 8 greatly reduced, but also the method proposed in this paper can quickly track the control signal of the upper manipulator servo system during the starting process, ensuring the overall control accuracy.

Then, when the disturbance occurs, we apply high-frequency vibration noise with different amplitudes to the manipulator to simulate the mechanical vibration noise in the actual application process. As shown in Figure 12, the method in [14] can be used to estimate this complex disturbance to a certain extent, but the details after the combination of the two disturbances are not accurately estimated, so there is a large estimation error, resulting in a large estimation error in the velocity signal of the estimated location.

However, using the method proposed in this paper, we can estimate the details of the combination of noise and disturbance. As shown in Figure 13, the influence of simulated mechanical vibration noise on the disturbed signal can be seen in the disturbance estimation, and the estimation error is also significantly reduced. At the same time, the advantages of the method proposed in this paper compared with the method proposed in [14] in the application of a manipulator controller can be seen in the position estimation error and speed estimation error. Additionally, the effectiveness and feasibility of the method proposed in this paper can be proven through experiments.

## 5. Conclusions

In this paper, we combine the BKS operator and the sliding mode controller of a robotic manipulator to establish a novel trajectory tracking control scheme for the robot manipulator servo system to solve the problem of low trajectory tracking accuracy under complex external perturbations. The BKS operator can accurately estimate the parameter update trend of the system and the observer based on the previous sampled data and the results of the perturbation observer; the method effectively reduces the steady-state error due to the perturbation estimation error. Introducing an estimation function effectively reduces the steady-state error of the servo control system due to the perturbation estimation error. The proposed control strategy retains the advantages of the traditional sliding mode controller while improving the tracking accuracy of the robotic servo system under complex perturbations. Finally, simulations and experiments comprehensively verify the effectiveness and feasibility of the proposed method.

The scheme proposed in this paper is applicable to scenes with complex disturbances and noises or to industrial tasks with high requirements for accuracy. However, in some scenes with low requirements for accuracy or a single disturbance, the calculation amount of the method proposed in this paper is larger than that of traditional methods and is not suitable for cases with requirements for running time. Therefore, in the future, we will develop a disturbance identification system to switch different control strategies in complex scenes where the disturbance form changes frequently. While improving the control accuracy, we also need to ensure the running speed of the robotic manipulator so that it can be applied to a wider range of application cases.

## Figures and Tables

**Figure 1 micromachines-14-00044-f001:**
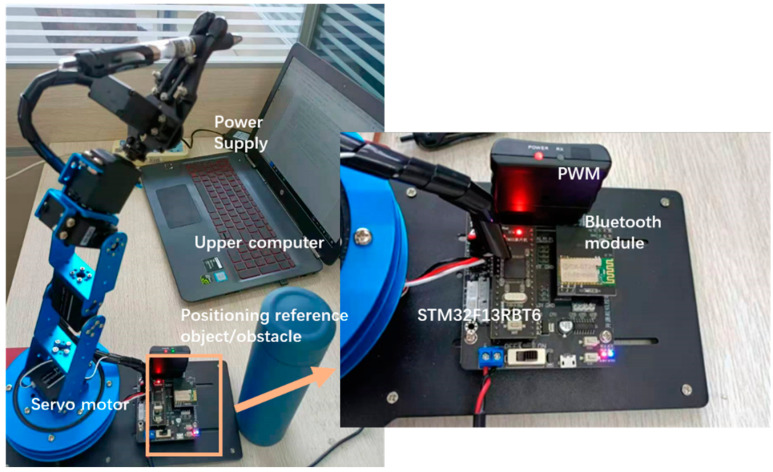
Manipulator experiment installation.

**Figure 2 micromachines-14-00044-f002:**
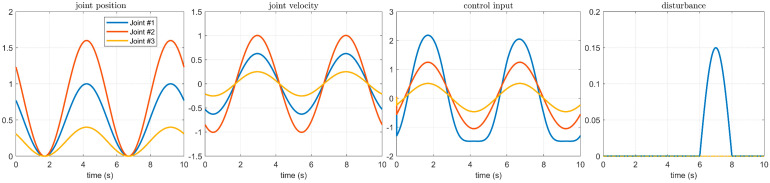
Manipulator motion trajectory, controller input, and disturbance.

**Figure 3 micromachines-14-00044-f003:**
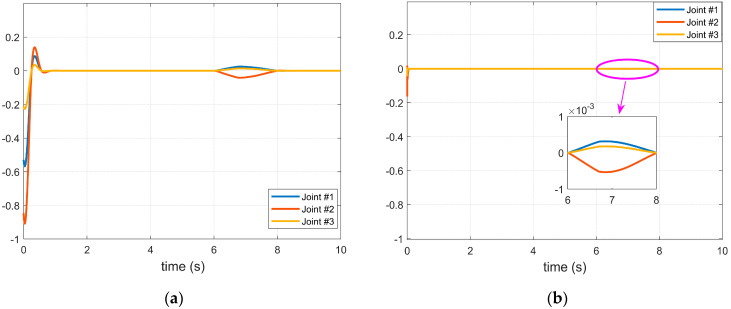
Effect of disturbance on servo motor of each axis of manipulator: (**a**) The BKS operator is not applied; (**b**) The BKS operator is applied.

**Figure 4 micromachines-14-00044-f004:**
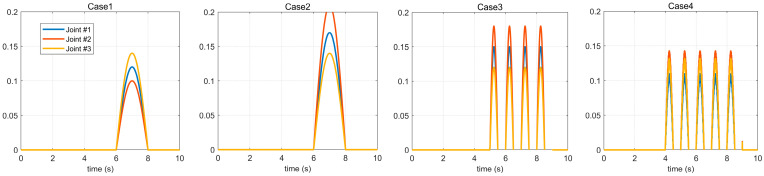
External disturbances in the four cases.

**Figure 5 micromachines-14-00044-f005:**
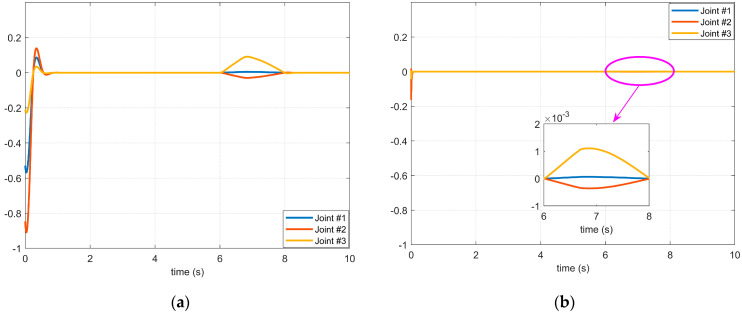
The effect of the 1st disturbance on the servo motor of each axis of the manipulator: (**a**) The BKS operator is not applied. (**b**) The BKS operator is applied.

**Figure 6 micromachines-14-00044-f006:**
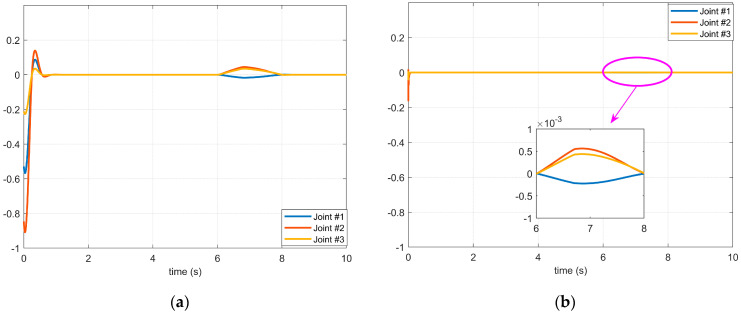
The effect of the 2nd disturbance on the servo motor of each axis of the manipulator: (**a**) The BKS operator is not applied. (**b**) The BKS operator is applied.

**Figure 7 micromachines-14-00044-f007:**
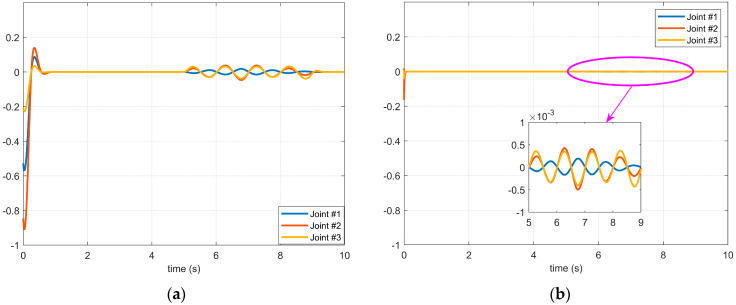
The effect of the 3rd disturbance on the servo motor of each axis of the manipulator: (**a**) The BKS operator is not applied. (**b**) The BKS operator is applied.

**Figure 8 micromachines-14-00044-f008:**
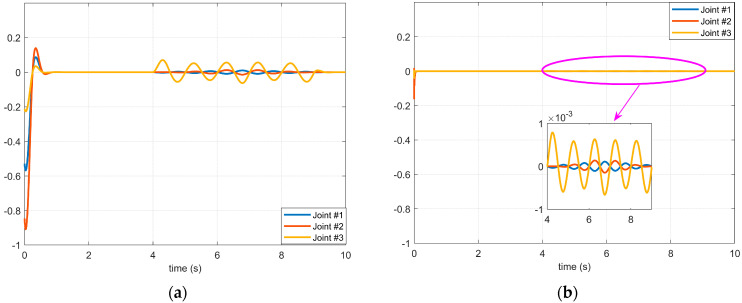
The effect of the 4th disturbance on the servo motor of each axis of the manipulator: (**a**) The BKS operator is not applied. (**b**) The BKS operator is applied.

**Figure 9 micromachines-14-00044-f009:**
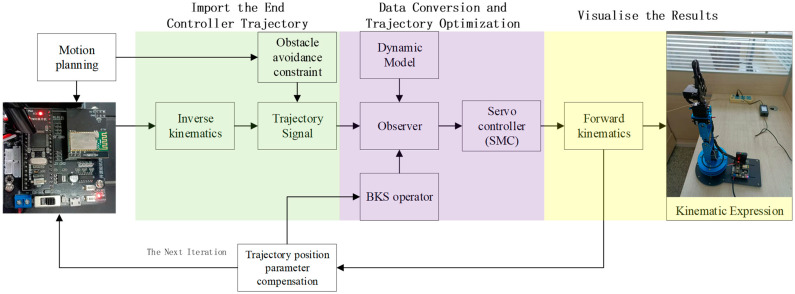
The control block of the robot manipulator servo control scheme proposed in this paper.

**Figure 10 micromachines-14-00044-f010:**
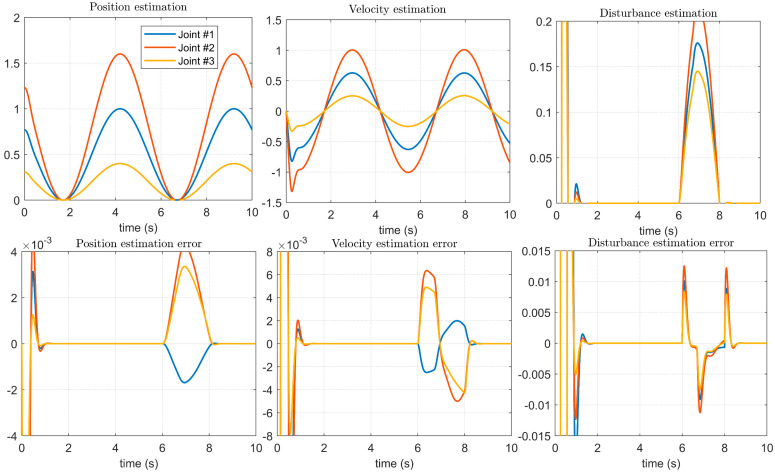
Estimation of control signals and disturbances using the method in [14].

**Figure 11 micromachines-14-00044-f011:**
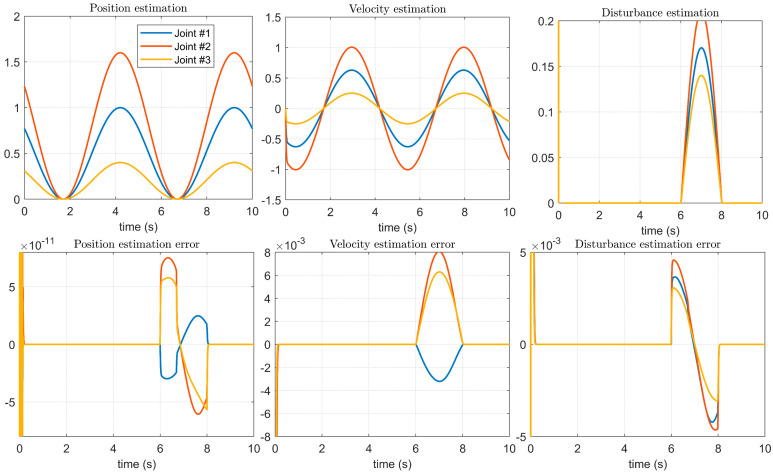
Estimation of control signals and disturbances using the proposed method.

**Figure 12 micromachines-14-00044-f012:**
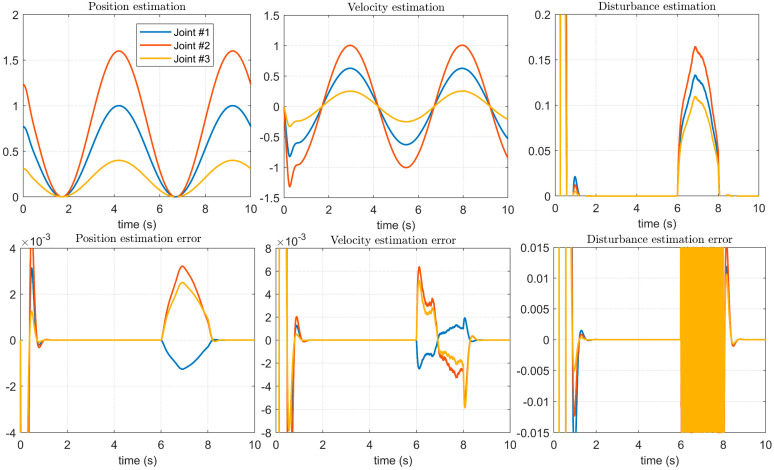
Estimation of control signals and disturbances with noise using the method in [14].

**Figure 13 micromachines-14-00044-f013:**
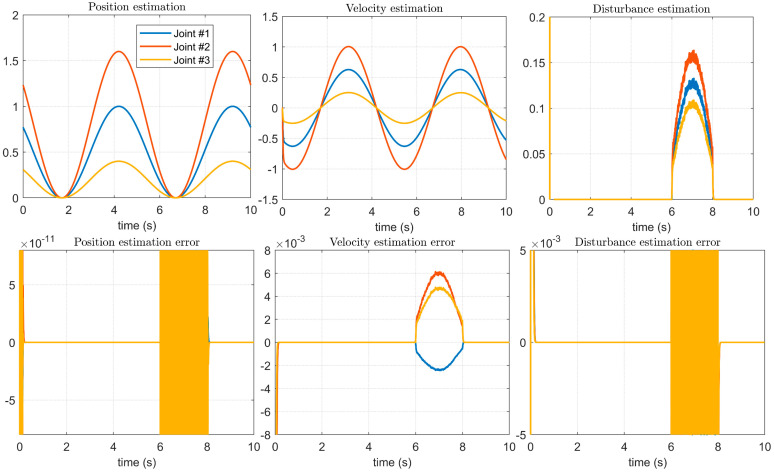
Estimation of control signals and disturbances with noise using the proposed method.

**Table 1 micromachines-14-00044-t001:** Maximum absolute error of approximation for the function $\hat{D}\left(t\right)$.

Type of Operator	n = 6	n = 40
BK	0.89441	0.31056
q-BK	1.08117	1.07137
α-BK	0.80731	0.30193
λ-BK	0.87743	0.30175
BKS	0.71316	0.24901

**Table 2 micromachines-14-00044-t002:** Parameters of the controller in simulation.

Parameters	Value
$K_{p}$	$14*I_{n}$
$K_{d}$	$10*I_{n}$
$\mu_{d}$	5
$k_{d}$	10
$l_{d}$	2/3
$\lambda_{i}$ *	(−2.0, 4, 5.5, 4, −2.0)

* Parameter values when n=6.

## Data Availability

Not applicable.

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
