# Peer review of "A Modified Robotic Manipulator Controller Based on Bernstein-Kantorovich-Stancu Operator"

_micromachines, 2022, doi:10.3390/mi14010044_

Round 1
Reviewer 1 Report
This manuscript presents an improved BKS operator which simplifies the controller parameters, reduces the algorithm complexity, and suppresses external interference and noise. By combining the disturbance observer with the new BKS operator, the recognition accuracy of the manipulator controller is effectively improved. Dynamic model of the manipulator is established, and the trajectory tracking scheme is improved by designing the controller. The improvement of the observer helps solve some real-time control problems in manipulator application. Simulated and experimental results valid the present method well.
Here are some further advices: 1. I suggest to place the physical map of the key parts at the top of the article to give the audience an initial idea of the basic model.
2. Some essential schematics would be needed. Otherwise, it is difficult for the audience to understand the meaning of each parameter when reading forward and back among formulas and the text.
3. Novelty claim should not exist in the tile and abstract.
4. An appropriate video demo would be really helpful for the audience to understand the superiority of your work.
5. The authors need to phrase some text to explain the application scenario of the present improved algorithm especially on the limits with respect to different control challenges in reality.
Reviewer 2 Report
I have reviewed the paper in detail and spend a lot of time to do this. The athors improve the manipulator controller by implementing Bernstein-Kantorovich-Stancu operator to their model. They propose a robotic manipulator controller to improve the error tracking accuracy of the manipulator controller when observing complex disturbances and noises.
The dynamic equation of n-DOF manipulator under external disturbance is given by following equation:
?(?)?̈ + ?(?, ?̇ )?̇ + ?(?) + ?(?̇ ) = ? + ?d, |
where M, C, G, and F are matrices.
The dynamic equation that express the uncertainty of the manipulator is given below:
?̈ = ?0(?, ?̇ ) + ?0(?)-1?(?) + ?(?),
where D represents the error caused by system uncertainty and disturbance to the joint motion, which can also be regarded as the compensation of the manipulator to suppress the uncertainty. ? is a bounded and known nonlinear function obtained from the manipulator model.
1. Line 155-156: The sentence "Assumption in the system there is a continuous function ?(?), which satisfies ? ⊆ 155 ?n as positive definite function, and can satisfy the following conditions:" is not understandable, please revise it.
2. Line 87: It is very important for this paper to give a comprehensive literature about Bernstein Kantorovich type operators to give an idea for the possible future works. So the sentence "such as α-BK [17], ?-BK [18]" should be revised as "such as ?-BK [a,b] α-BK [17], ?-BK [18]"; a=Mathematics, 9(16), (2021) 1895. b=Mathematics, 10(12) (2022) 2027.
3. Line 88: "other methods have also been developed" should be "other methods have also been developed [a,b,c,d]" a=Mathematics 10(7) (2022) 1149. b=Numerical and theoretical approximation results for Schurer–Stancu operators with shape parameter $\lambda$, Comp. Appl. Math. 41 (2022) 181. c=Approximation by (p, q) $(p, q) $-LupaÅŸ–Schurer–Kantorovich operators, K Kanat, M SofyalıoÄŸlu - Journal of Inequalities and Applications, 2018. d=On Stancu type generalization of (p, q)-Baskakov-Kantorovich operators, K Kanat, M SofyalıoÄŸlu - Communications Faculty of Sciences University of, 2019.
4. Lines 88-89: "In this paper, a new BKS [19] operator is introduced into the manipulator controller..." should be "In this paper, a new BKS operator, which is introduced in [19], is implemented to the manipulator controller...".
Subsection 2.2 is about certain Bernstein-Kantorovich-Stancu operator that was first defined in A numerical comparative study of generalized Bernstein-Kantorovich operators. Math. Found. Comput. (2021), 4, 311. Then this paper was extended to Bernstein-Kantorovich-Stancu operator.
5. So
a) the original study in "Math. Found. Comput." must be given at the begining of subsection 2.2.
b) please revise "we define the BKS operator applied to the manipulator" as "we apply the BKS operator to the manipulator" (Line 165).
c) Line 165: Please revise the sentence "The modified BKS operator is defined as" as "The modified BKS operator is defined in [19] as".
d) please give the citations for the expressions in Eq. 10 and Eq. 11.
e) Line 179: "we can obtain" should be "the following moments were obtained in [19]".
f) please explain how/when/where do you need moments that are given in Eq. 13. And what are their roles in your model? This is very important.
g) Lines 183-184: The sentences "In addition, we can write as follows" and "This means that we can prove that" must be revised as "In addition, following results were obtained in [19]:"
6. Lines 242-246 sould be revised, it is complicated.
7. Please make it very clear that how do you obtain Eq. (34) (according to the design of the controller) from the original definition of Berntein-Kantorovich in Eq. (9). Please a short proof even if it is clear for you.
8. Eq. 38 and Eq 39. Please mention that Wf and Zf are matrices. Also write these matrices in the matrix form (there should be some space between terms of a matrix).
9. Eq. 39: Please correct the expression "t(m+p,?-\alpha) ".
10. Eq. 40: Please express "?D , ?D and ?D explicitly.
11. Line 260: Please explain what is "BKS Operator weight approximation error".
12. Table 1: Please make it clear that which D(t) function did you use to get your results.
13. The BKS operator that you use in your paper includes the parameter ?i. So what are the values of parameters ?i in your numerical/simulation results? It is not written.
14. Which computer programs did you use to get your numerical/simulation results?
The organization of the paper is good. The results are original. The authors should be very careful about all my remarks. After a careful revision the paper can be accepted. I want to see the revised version of the paper.
Round 2
Reviewer 2 Report
I thank the authors for considering my corrections. They corrected/revised the paper very well. I have one small remark: please remove Equation numbers when you do not use them. The quality of the paper is very good. The current version is ready for acceptance.